# Lightweight Image Classification Network Based on Feature Extraction Network SimpleResUNet and Attention

## Abstract

Using lightweight neural network models for small-sample image classification tasks has always been a challenging task. This paper proposes a feature extraction network called SimpleResUNet based on ResNet and U-Net, and adopts the Attention mechanism as the feature classifier for image classification, aiming to improve the accuracy and robustness of small-sample image classification tasks using lightweight network structures. Firstly, the network combines the feature extraction capability of U-Net and the efficient feature propagation capability of ResNet to effectively extract details and contextual information from the images. Secondly, the Attention mechanism is used to capture the correlations and dependencies between different features in the feature sequence. Multiple public datasets were used for verification in the experiment, and comparative analysis was conducted with other methods. Experimental results show that the network achieves superior performance in image classification tasks. Finally, some thoughts on the mechanism of this model are discussed, the work of this paper is summarized, and future research directions are prospected.

## 1 Introduction

Image classification is an important task in the field of computer vision, which plays a key role in many practical applications, such as medical image analysis, autonomous driving, security monitoring, etc. With the development of deep learning, deep convolutional neural networks (CNN) have achieved remarkable success in image classification tasks. However, the traditional CNN model still has certain limitations, such as local receptive field limitations, large amounts of annotated data, large parameters, and high computational complexity. In particular, the convolutional layer of CNN extracts local features of the image through local receptive fields, which limits the network's ability to understand global information to a certain extent. For some image classification tasks that require global contextual information, such as identifying scenes in images or long-distance dependencies in images, CNN may not be able to fully capture global features.

To overcome these challenges, researchers have proposed many improved network structures and techniques. Among them, residual connection and attention mechanism (Bahdanau et al. (2014)) are two important techniques that are widely used in image classification tasks. Residual U-net combines the advantages of residual connections and U-net (Ronneberger et al. (2015)) structure, can effectively extract details and contextual information in images, and achieves excellent performance in image segmentation tasks.

U-net is a classic image segmentation network with an encoder-decoder structure and skip connections. It improves the performance of the segmentation task by concatenating the feature maps of the encoder and decoder, enabling the network to utilize feature information at different levels simultaneously. Residual connections (He et al. (2016)) are a technique that adds the input directly to the output of the network through skip connections. It can effectively solve the gradient disappearance and gradient explosion problems in deep networks, make the network easier to train, and improve the transfer ability of features. The residual connection in the U-net network allows information to be skip-connected in the network, combining low-level features at the bottom with high-level semantic features at the top. This connection method helps to solve the problem of gradient dis-

appearance and information loss in the deep network, so that the network can better transmit and retain the detailed information in the image. The structure of U-Net is a variant of the AutoEncoder, which transfers features through skip connections between the encoder and decoder. Although skip connections help preserve low-level and high-level features, U-Net lacks an explicit mechanism to capture long-distance contextual information.

The Attention mechanism can automatically learn and adjust attention weights based on different image feature sequences, allowing the network to pay more attention to feature sequences that are more important to the current task and suppress features that are irrelevant to the task. The adaptive attention mechanism can improve the network's perception of key features, thereby improving the ability to express and distinguish image features.

The contributions of this paper mainly include:

1) Proposed a lightweight network structure suitable for small samples, with only 2.50M parameters, lower requirements in terms of memory usage and computing resources, and faster inference speed. A new Residual U-net structure SimpleResUNet is proposed, which can combine the characteristics of U-net that can pay attention to global and local features at the same time and the characteristics of good information transfer of residual network, and apply this structure to image classification tasks. The image feature attention classification network is introduced to change the focus of the Attention mechanism from the image itself to the superimposed image features, and GroupNorm is used to improve the accuracy and robustness of classification when there are insufficient samples.

2) The gradient calculation formula during the back propagation process of the SimpleResUNet network structure is given, which proves that this structure inherits the gradient calculation advantages of ResNet and has good gradient propagation efficiency.

3) Traditional image classification networks are suitable for image classification tasks with fixed image sizes. This paper extends this model to multi-scale image classification tasks by introducing an adaptive tie pooling layer.

4) The interpretability of the model is discussed, and an inference is proposed about using signal processing related theories to explain the feature space dimensions of the model.

## 2 RELATED WORK

The related work of this article mainly includes three parts: lightweight network structure, fusion network of U-net and ResNet, and attention mechanism.

### 2.1 LIGHTWEIGHT NETWORK STRUCTURE

SqueezeNet (Iandola et al. (2016)) was the first lightweight convolutional neural network model proposed, which has the same accuracy as AlexNetSimonyan & Zisserman (2014). MobileNet (Howard et al. (2017), Sandler et al. (2018), Howard et al. (2019)) combines the structural ideas of ResNet(He et al. (2016)) with NAS search, and uses depth-separable convolution, which greatly improves computing efficiency. ShuffleNet (Zhang et al. (2018), Ma et al. (2018)) uses the channel shuffle operation to make up for the information exchange between groups, which can reduce the main network calculation amount and increase the dimension of convolution. Xception (Chollet (2017)) interprets the Inception module in convolutional neural networks as an intermediate step between conventional convolution and depth-separable convolution operations. It proposes a convolutional neural network architecture based entirely on depthwise separable convolutional layers, with a linear stack of depthwise separable convolutional layers with residual connections. These models reduce model complexity to varying degrees while reducing model accuracy.

### 2.2 U-NET AND RESNET

LinkNet (Chaurasia & Culurciello (2017)) introduces a residual connection into the original UNet, and directly connects the encoder to the decoder to improve accuracy and reduce processing time to a certain extent. D-LinkNet (Zhou et al. (2018)) uses LinkNet as the basic skeleton, uses the trained ResNet on the ImageNet dataset as the encoder of the network, and adds a dilated-convolution layer with a shortcut in the center to make the entire network more capable of recognition. The

receiving field is larger and multi-scale information is integrated. Res-UNet (Xiao et al. (2018)) and Dense-UNet (Guan et al. (2019)) replace each sub-module of UNet with a form with residual connection and dense connection respectively. UNet3+ (Huang et al. (2020)) believes that although UNet++ (Jha et al. (2019)) uses dense skip connections, it does not make full use of multi-scale information, so UNet3+ proposes full-scale skip connections. DC-UNet (Lou et al. (2021)) uses DC block to replace the MultiRes block in MultiResUNet. TransUNet (Chen et al. (2021)) did not use Transformer to completely replace the CNN in UNet, but used Transformer as a middleware to complete long-distance dependency modeling. These models increase the depth and number of parameters in the network, resulting in the need for more computational resources and time during training and inference. In this paper, we draw inspiration from the structure of U-Net and combine it with ResNet to build a novel Residual U-Net architecture. We employ this architecture for feature extraction in image classification tasks, aiming to enrich the feature space of the images.

### 2.3 ATTENTION

The attention mechanism is a technology that can adaptively learn and focus on the most informative areas in an image. By introducing the attention mechanism, the network can automatically learn and focus on important features, improving the accuracy and robustness of classification. Transformer (Vaswani et al. (2017)) abandoned the traditional CNN and RNN, and introduced self-attention mechanism (self-attention). The self-attention mechanism can calculate the dependencies between positions in the input sequence and weight each position according to these dependencies. Retnet (Sun et al. (2023)) is similar to Transformer, stacking the same modules of the L layer, and each module contains two sub-modules of multi-scale retention (MSR) and feed-forward network (FFN). In this article, we use the Attention mechanism as a classifier and connect it with the image feature space extracted by SimpleResUNet, which can capture the associations and dependencies between different features in the image and improve the performance of the network in image classification tasks.

## 3 METHOD

The image classification network proposed in this paper consists of three parts: feature extraction network, feature scale normalization network and attention classification network, as shown in Figure 1. The feature extraction network is a SimpleResUNet, which continuously superimposes features in the image through residual learning. The feature scale normalization network is composed of a fully convolutional network and an adaptive average pooling layer, which normalizes the scale of the features extracted by the feature extraction network, so that the network can adapt to the training of images of all scales. The attention detection network classifies scale-normalized features through the attention maintenance mechanism.

The U-net network adopts a symmetrical upsampling and downsampling structure. The downsampling operation can gradually expand the receptive field and extract higher-level semantic information, while the upsampling operation can restore the spatial resolution of the image and retain detailed information. This structure enables the network to focus on global and local features at the same time, leading to better feature extraction. SimpleResUNet inherits U-net's method of expanding the receptive field by fusing features at different levels to combine low-level detailed features with high-level semantic features. This multi-level feature fusion can provide a richer feature representation, enabling the network to better understand the structure and content of the image.

The Attention mechanism provides richer semantic information by capturing the context information in the image feature sequence and integrating global and local semantic information, thereby improving the discrimination and expressive ability of image features, which is especially important for image classification tasks. It adjusts attention weights, emphasizes important features, reduces dependence on irrelevant features, improves the discriminative ability and robustness of image features, and improves the accuracy and robustness of classification tasks. The attention map of the generated features at the same time shows the degree of attention of the network to the image feature sequence, provides an explanation and understanding of the network's decision-making, and helps analyze the behavior of the network and improve the interpretability of the model.

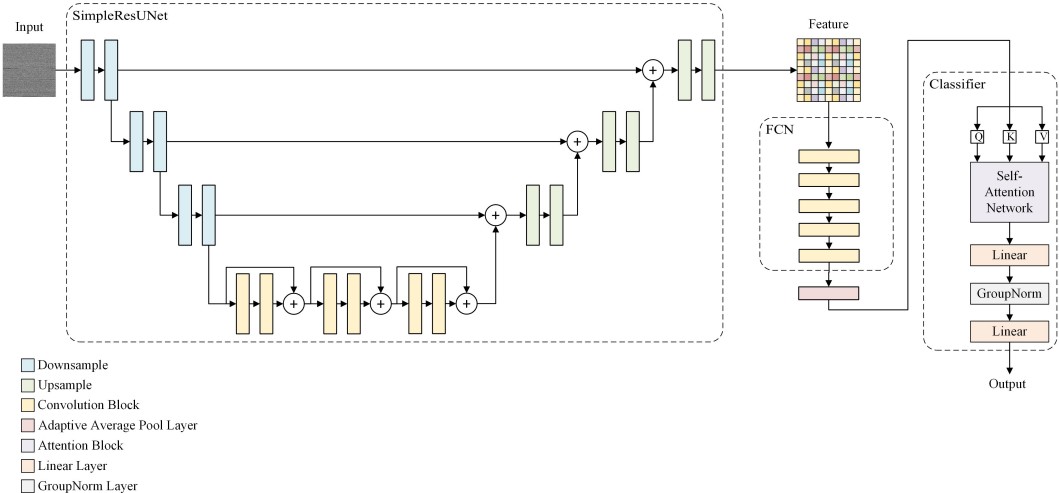

Figure 1: Overall network structure. Among them, the output of each layer of SimpleResUNet downsampling forms a residual structure with the network of the lower layer. The bottom layer network is composed of shallow ResNet, and the output of SimpleResUNet is a set of high-dimensional features. This set of high-dimensional features will be dimensionally reduced to a unified feature dimension through the FCN network and adaptive average pooling layer to form a feature sequence. This set of feature sequences will be input to a Self-Attention classifier using GroupNorm.

This article compares the number of parameters (Params) and floating point operations (FLOPs) of this model with existing models. The results are shown in Table 1. The number of parameters of this model is much smaller than that of traditional image classification networks, and is similar to the number of parameters and floating-point operations of existing lightweight neural network structures.

Table 1: Comparison of parameters between this model and existing neural network classification models.

| Model | Params(M) | FLOPs(G) | Model | Params(M) | FLOPs(G) |
|---|---|---|---|---|---|
| AlexNet | 61.10 | 0.77 | MnasNet0_5 | 2.22 | 0.14 |
| VGG16 | 138.36 | 15.61 | MnasNet0_75 | 3.17 | 0.24 |
| VGG16_BN | 138.37 | 15.66 | MnasNet1_0 | 4.38 | 0.34 |
| VGG19 | 143.67 | 19.77 | MnasNet1_3 | 6.28 | 0.53 |
| VGG19_BN | 143.68 | 19.83 | MobileNet_v2 | 3.50 | 0.33 |
| ResNet18 | 11.69 | 1.82 | ShuffleNet_v2_x0_5 | 1.37 | 0.05 |
| ResNet50 | 25.56 | 4.14 | ShuffleNet_v2_x1_0 | 2.28 | 0.15 |
| DenseNet121 | 7.98 | 2.90 | ShuffleNet_v2_x1_5 | 3.50 | 0.31 |
| SqueezeNet1_0 | 1.25 | 0.82 | ShuffleNet_v2_x2_0 | 7.39 | 0.60 |
| SqueezeNet1_1 | 1.24 | 0.35 | inception_v3 | 27.16 | 5.75 |
| Ours | 2.50 | 0.586 | | | |

## 3.1 BACKPROPAGATION OF SIMPLERESUNET

A neural network layer consists of a set of perceptrons. Each perceptron uses an activation function to map a set of inputs to output values. Neural network functions are formed in chains:

$$f_k(x) = f^{(k)}(...f^{(2)}(f^{(1)}(x)))$$

Among them, $f^{(i)}$ is the function of the $i$-th layer of the network, $i = 1, 2, ..., k$, and k is the number of layers of the network.

For a residual network, each residual block can be represented as:

$$R^{(l)}(x) = R(x) + x$$

A complete residual network consists of multiple residual blocks and can be expressed as

$$R_l(x) = R^{(l)}(...R^{(2)}(R^{(1)}(x)))$$

For any set of continuous network layers, it can be abstracted as a function $F(x)$, which allows $F(x) = R_l(x)$. Then in the deep residual U-net network, any pair of upsampling and downsampling combinations can be expressed as:

$$H(x) = F(x) + x$$

That is, the input $x_{l+1} = x_l + F(x_l)$ corresponds to any downsampling layer, then when recursing to the $L$th layer, there is

$$x_L = x_l + \sum_{i=l}^{L} F(x_i)$$

Then, the backpropagation in this case is

$$\frac{\partial \varepsilon}{\partial x_l} = \frac{\partial \varepsilon}{\partial x_L} \frac{\partial x_L}{\partial x_l} = \frac{\partial \varepsilon}{\partial x_L} \left( 1 + \frac{\partial}{\partial x_l} \sum_{i=l}^{L} F_i(x_i) \right)$$

In this way, the gradient of each layer of upsampling can include the gradient of the previous upsampling layer and the gradient of the corresponding downsampling layer, which not only avoids the disappearance of gradients, but also makes the network easier to train.

### 3.1.1 SELF-ATTENTION CLASSIFIER

Self-Attention is an attention mechanism used in the model, which captures the relationship and importance between elements through a weighted representation of other elements in the sequence when processing each element in the sequence.

Suppose there is an input sequence $X$, which contains $N$ elements, and the representation of each element is $x_i$, where $i$ represents the index of the element, $i \in [1, N]$. First, the input sequence needs to be mapped to three different representation spaces through linear transformation: query, key and value.

Through linear transformation, three representation matrices $Q$, $K$ and $V$ are obtained, and their dimensions are $d_q$, $d_k$ and $d_v$ respectively. where $d_q$, $d_k$ and $d_v$ are the dimensions of the query, key and value.

For the input sequence $X$, we can obtain the query matrix $Q$, key matrix $K$ and value matrix $V$ by the following linear transformation:

$$Q = XW_q$$

$$K = XW_k$$

$$V = XW_v$$

Where $W_q$, $W_k$ and $W_v$ are learnable weight matrices.

Attention weights are used to measure the correlation between each element and other elements, while the weighted representation is obtained by multiplying the attention weights with the value matrix. First, calculate the similarity score between the query matrix Q and the key matrix K, use the dot product to calculate the similarity and multiply it with the value matrix $V$ to get a weighted representation:

$$O = (QK^T \odot D)V$$

Among them, $D$ is the scaling factor used to scale the result of the dot product.

$$D = \frac{1}{\sqrt{d_k}}$$

## 3.2 GROUPNORM LAYER

In the attention network, the traditional LayerNorm (Ba et al. (2016)) layer is replaced by the Group-Norm (Wu & He (2018)) layer. GroupNorm divides the features into several groups, and the features in each group are independently normalized. This approach works well for smaller batch sizes and smaller feature dimensions, and is computed as

$$\hat{x}_i = \frac{1}{\sigma_i}(x_i - \mu_i)$$

$$\mu_i = \frac{1}{m} \sum_{k \in S_i} x_k, \sigma_i = \sqrt{\frac{1}{m} \sum_{k \in S_i} (x_k - \mu_i)^2 + \epsilon}$$

$$S_i = \{k | k_N = i_N, \left\lfloor \frac{k_C}{C/G} \right\rfloor = \left\lfloor \frac{i_C}{C/G} \right\rfloor\}$$

Compared to BatchNorm (Ioffe & Szegedy (2015)), GroupNorm has lower batch size requirements because it normalizes within each sample and does not rely on statistics between samples within a batch. This allows GroupNorm to work more stably with smaller batch sizes. LayerNorm normalizes features within each sample independent of batch and spatial dimensions, which makes LayerNorm suitable for smaller batch sizes and larger feature dimensions.

## 3.3 ADAPTIVE AVERAGE POOLING LAYER

The adaptive pooling layer can adjust the size and stride of the pooling window according to the size of the input data and the feature distribution, so as to realize the extraction of features of different scales. In image tasks, the size and shape of objects may change. Using adaptive pooling layers can better capture feature information at different scales and improve the network's ability to recognize objects of different scales.

Traditional pooling layers use fixed pooling window sizes and strides for downsampling, which may lead to information loss. Especially when the size of the input data does not match the fixed pooling window, some information will be discarded. The adaptive pooling layer can automatically adjust the size and stride of the pooling window according to the size of the input data, making the pooling operation more accurate, reducing the loss of information, and helping to improve the performance of the network. The adaptive pooling layer can dynamically adjust the size and stride of the pooling window according to the size and feature distribution of the input data, thereby reducing the number of parameters in the network. Compared with the fixed pooling window size and stride, the adaptive pooling layer can perform flexible downsampling operations according to the characteristics of the input data, reducing the complexity of the network and reducing the risk of overfitting.

Suppose the input feature map is X, its size is $C \times H \times W$, where C is the number of channels, H is the height, and W is the width. Suppose we want the output to be of size $H'W'$, where H' is the target height and W' is the target width.

The operation of the adaptive average pooling layer can be divided into the following steps: (a) Calculate the size of the input feature map: the size of the input feature map is $C \times H \times W$. (b) Calculate the size of the pooling area: According to the target height and width, calculate the size of the pooling area as $H_{pool} = H/H'$ and $W_{pool} = W/W'$. (c) Divide the pooling area: Divide the input feature map into a pooling area of size $H_{pool} \times W_{pool}$. (d) Calculate the pooling output: For each pooling area, calculate its average value and use it as the value of the corresponding position of the output.

The size of the output feature map Y is $C \times H' \times W'$, where the value of each position $(c, h', w')$ of Y is:

$$Y(c, h', w') = \frac{1}{(H_{pool} \times W_{pool})} \sum_{i \in h_{pool}, j \in w_{pool}} X(c, h_{i,j}, w_{i,j})$$

Among them, $(h_{pool}, w_{pool})$ represents the position of the pooling area, and the range is $(h' \times H_{pool}, (h' + 1) \times H_{pool})$ and $(w' \times W_{pool}, (w' + 1) \times W_{pool})$.

## 4 EXPERIMENT

This article adopts the Python experimental environment, uses the PyTorch deep learning framework, and provides computing support through NVIDIA Geforce 1050 GPU. The implementation is divided into three stages, which are data set preparation, model training and evaluation stage, and experimental result analysis stage. Among them, the data set preparation stage needs to prepare a variety of different image data sets to verify the accuracy of the model on different data sets. In the model training and evaluation stage, the experimental data is input into the built network for training, and the corresponding loss value and accuracy rate are obtained. In the experimental result analysis stage, the model training results are compared with other related model results, and the pros and cons of each index are analyzed.

### 4.1 DATASETS

A variety of image classification datasets from different fields are used in the experiments, including the general image classification dataset CIFAR-10 (Krizhevsky et al. (2009)), and the malware image datasets MalImg (Gibert et al. (2019)) and MalVis (JIANG & QIN (2023)).

The CIFAR-10 dataset was created by Jeffrey Dwork et al. in 2009, and contains 10 categories of color images respectively. The scale of this dataset is relatively large, with a total of 160,000 images. The CIFAR-10 dataset is widely used to test tasks such as object recognition and scene recognition.

The MalImg dataset is a commonly used dataset for malicious code image classification, which contains 9,427 image samples from 25 malware families and 4,321 image samples from 9 benign software families, for a total of 13,748 image samples. Each image in the MalImg dataset is represented as a grayscale image with a size of 32x32 pixels. These images are extracted from malicious code samples, preprocessed and transformed for image classification tasks.

The MalVis dataset aims to provide researchers with a real-world RGB-based dataset for evaluating vision-based multi-category malware recognition research. The dataset contains byte images of 26 categories, of which 1 category represents a "legitimate" sample and the remaining 25 categories correspond to different malware types. The data set includes 9100 training images and 5126 verification images. There are 350 samples in each category in the training set, and a large number of "legitimate" samples (1482) in the verification set for malware discrimination.

## 4.2 MODEL TRAINING AND EVALUATION

To show the performance of the proposed method, several evaluation metrics are used. These metrics include accuracy, precision, recall, and f-score. The performance index calculation formula is as follows:

$$Accuracy = \frac{TP + TN}{TP + FP + FN + TN}$$

$$Precision = \frac{TP}{TP + FP}$$

$$Recall = \frac{TP}{TP + FN}$$

$$F - score = \frac{2TP}{2TP + FP + FN}$$

Here, TP refers to true positive, FP is false positive, and TN is true negative, TP is true positive.

In the experiment, it was found that the recognition accuracy of the model is related to the output dimension of the extracted features of the Residual U-net network, which is collectively referred to as the feature dimension below. As shown in Table 2, the model has a good performance on the MalImg dataset. When the feature dimension is 64, the accuracy reaches 99.31%. When the feature dimension is 256, it is not much different from this result. Precision, Recall and F-score are also much higher than other models. Table 2 presents the results of the model on the MalVis dataset, and the model outperforms other existing models on datasets with a small sample size.

Table 2: Quantitative results on the MalImg dataset.

| Model | Accuracy(%) | Precision(%) | Recall(%) | F-score(%) |
|-------|-------------|--------------|-----------|------------|
| AlexNet | 97.80 | 93.12 | 93.73 | 96.56 |
| VGGNet | 98.01 | 93.12 | 93.73 | 93.33 |
| ResNet50 | 97.84 | 92.70 | 93.00 | 92.44 |
| Xception | 97.43 | 92.20 | 91.83 | 91.62 |
| ShuffleNet | 98.28 | 93.80 | 94.45 | 94.04 |
| MobileNet | 97.77 | 92.80 | 93.07 | 92.60 |
| Ours-3 | 96.78 | 95.04 | 96.23 | 96.13 |
| Ours-64 | 99.31 | 99.22 | 99.12 | 99.17 |
| Ours-256 | 99.60 | 99.34 | 99.28 | 99.29 |

Table 3 shows the performance of the model on the CIFAR-10 datasets, as well as the results of the model with different feature space dimensions. When the feature dimension is 64, the effect of the model is basically the same as the existing model, but when the feature dimension is 256, the effect of the model is better than the existing model.

## 4.3 ANALYSIS OF RESULTS

The dimensionality of features determines the richness of features that the network can express. If the feature dimension is low, the network may not be able to capture subtle differences and important features in the image, resulting in poor classification performance. Therefore, a higher feature dimension can usually provide richer feature expression capabilities and help improve the performance of classification learning. The higher the dimensionality of features, the correlation and redundancy between features may increase. Features in a high-dimensional feature space may contain redundant information, which may cause the classifier to overfit the training data, thereby reducing the generalization ability of the model.

Table 3: Quantitative results on the MalVis dataset.

| Model | Accuracy(%) | Precision(%) | Recall(%) | F-score(%) |
|---|---|---|---|---|
| AlexNet(Krizhevsky et al. (2012)) | 82.86 | 84.10 | 93.33 | 96.44 |
| VGGNet(Simonyan & Zisserman (2014)) | 95.14 | 90.46 | 91.07 | 90.62 |
| ResNet-50(He et al. (2016)) | 91.24 | 85.30 | 85.60 | 85.08 |
| Xception(Chollet (2017)) | 89.74 | 81.82 | 84.71 | 80.00 |
| ShuffleNet(Ma et al. (2018)) | 97.00 | 94.73 | 94.20 | 94.00 |
| MobileNet)Howard et al. (2019)) | 97.13 | 94.00 | 94.90 | 94.30 |
| Ours-64 | 98.05 | 98.06 | 98.05 | 98.05 |

In image classification tasks with smaller sample sizes and fewer categories, the model requires fewer feature dimensions. Table 4 shows the performance results of models with different feature dimensions in the same data set. The results show that the feature dimension of dimension 64 is sufficient for classification tasks with a small sample size. When training with larger data sets, the feature dimensions need to be appropriately adjusted according to the data set conditions.

Table 4: Quantitative results for different feature dimensions.

| Dimensions | Accuracy(%) | Loss | Precision(%) | Recall(%) | F-score(%) |
|---|---|---|---|---|---|
| 3 | 98.49 | 12.86 | 96.46 | 96.33 | 96.38 |
| 32 | 98.80 | 9.52 | 97.40 | 97.38 | 97.39 |
| 64 | 99.31 | 7.96 | 99.22 | 99.12 | 99.17 |
| 256 | 99.25 | 6.93 | 99.19 | 99.15 | 99.17 |

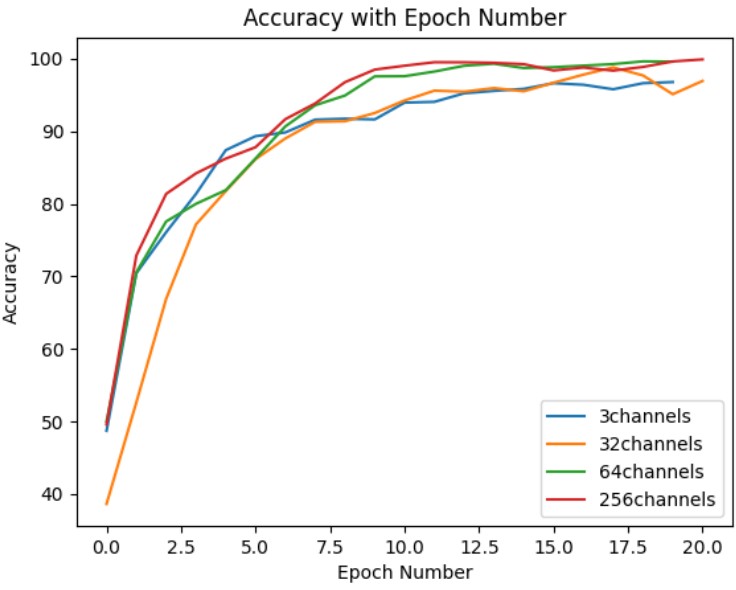

Figure 2: The impact of different feature dimensions on the classification accuracy of this model.

Figure 2 shows the accuracy change curve of training the same data set under different feature dimensions. It can be seen from the figure that the accuracy rate of the model with a feature dimension of 3 is the worst. With the increase of the feature dimension, the accuracy rate gradually increases,

and the feature dimension is not much different between 64 and 256, which proves that the feature space dimension required by the data set has been reached.

# 5 DISCUSSION

Figure 3 shows the loss change trend of training the same data set under different feature dimensions. It can be seen that the model with a feature dimension of 64 has the smoothest loss change trend, while the model with a feature dimension of 256 has the smallest final loss value. The model with a feature dimension of 32 produces large fluctuations during the training process. This part of the fluctuation happens to correspond to the part of the curve with a feature dimension of 32 that fluctuates in Figure 2.

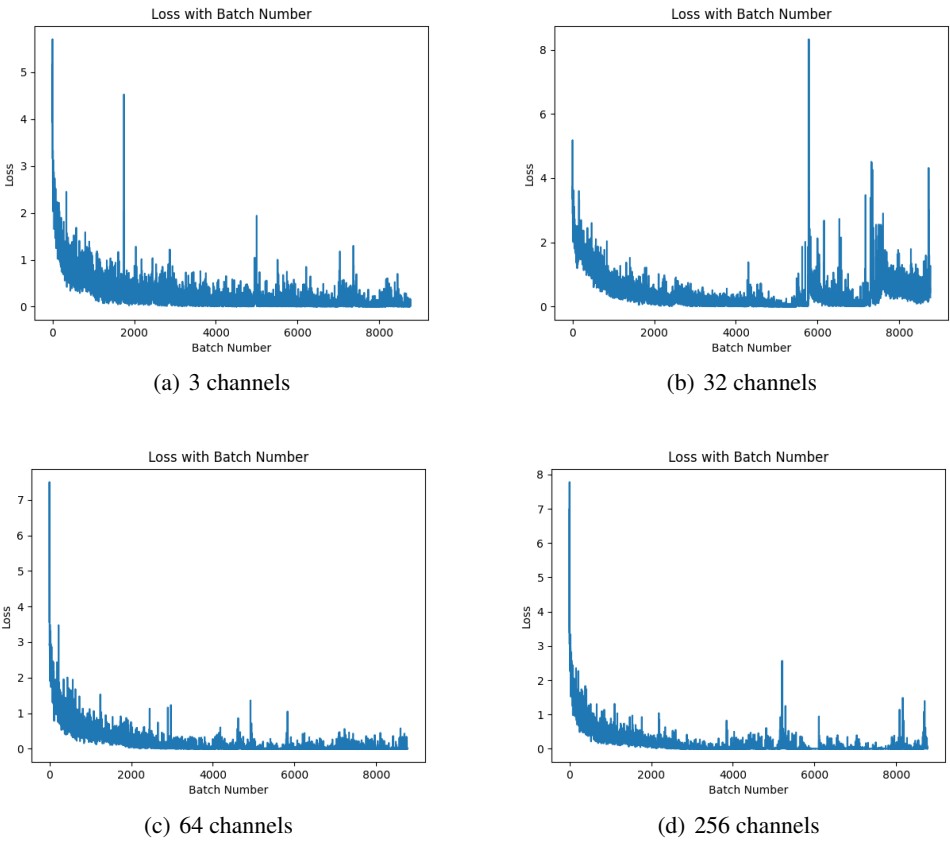

Figure 3: The impact of different feature dimensions on model learning loss values. (a) is the change trend of loss value when the feature dimension is 3. (b) is the change trend of loss value when the feature dimension is 32. (c) is the change trend of loss value when the feature dimension is 64. (d) is the change trend of the loss value when the feature dimension is 256.

We infer that the reason for this phenomenon may be related to sampling theory. During the network learning process, the continuous accumulation of features is similar to the superposition of signals of different frequencies layer by layer, as shown in Figure 4. The information carried by the signal is the extracted features, and the role of the classifier is like sampling in the feature space. Only by setting the sampling frequency to twice the maximum frequency can the information in the original features be completely retained. We assume that the total number of features has an upper limit. Increasing the feature space is equivalent to redistributing the features in the original feature space into the new feature space according to the original distribution rules.

Therefore, for a feature space dimension that can adapt to the data set, it should also be a value greater than or equal to twice the number of decisive features in the data set. If the dimension of

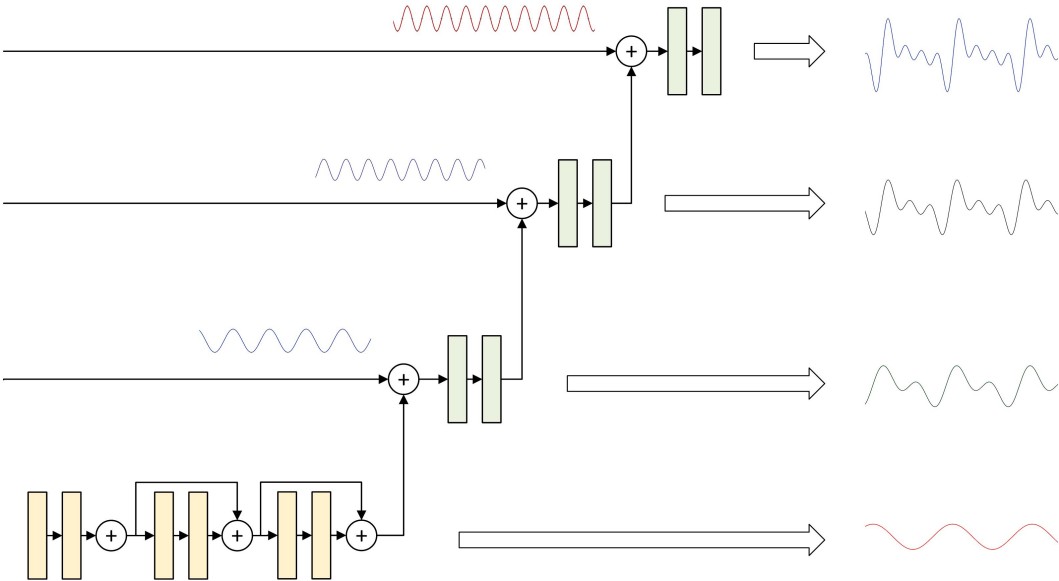

Figure 4: The feature accumulation process of this model can be seen as the continuous superposition of signals of different frequencies.

the feature space is less than twice the number of decisive features in the data set, spectral aliasing of different signals occurs in the feature space. Then the features in this feature space will continue to undergo large changes to adapt to the superposition of more decisive features, so there will be large fluctuations in the loss. And if the feature space dimension is greater than or equal to twice the number of decisive features in the data set, the feature space sampling result can fully retain the information in the original features. Then these features can stably exist in the feature space, and only minor adjustments will be made with the learning of the neural network, so that the final learning result of the neural network is very good.

## CONCLUSION

This paper proposes an image classification network based on Residual U-net and Attention, aiming to use lightweight neural network models to improve the accuracy and robustness of image classification tasks. By proposing a new Residual U-net network structure and Attention's adaptive attention mechanism, our network can effectively capture details and contextual information in images and perform image classification tasks through attention classifiers. The network was compared with other methods in multiple public data sets, and experimental results showed that the network showed superior performance in image classification tasks and showed good robustness. At the same time, some thoughts about the model are also discussed, which can provide new ideas for the follow-up work. Future research can further optimize the computing efficiency of the network, expand the applicable scope of the network, and further improve the performance of large-scale datasets.

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
