# OpenReview forum: "Lightweight Image Classification Network Based on Feature Extraction Network SimpleResUNet and Attention"
_ICLR.cc/2024/Conference — Submitted to ICLR 2024_

### Official Review · Reviewer_8gWN · 2023-10-28

**Soundness:** 3 good
**Presentation:** 2 fair
**Contribution:** 1 poor
**Rating:** 3
**Confidence:** 5

**Summary:**

This paper proposes a new image classification network composed of the ResUNet and a self-attention module. Experiments on malware image datasets show its advantage.

**Strengths:**

1. The writing is easy to understand.
2. The performance of the proposed method is greater than the compared networks.

**Weaknesses:**

1. The novelty is limited. The proposed SimpleResUNet and attention classifier are somehow very similar to previous works. For example, adopting residual blocks in the middle stage of UNet is a commonly used method for recent UNet-based methods, like StableDuffision. The contribution of the authors seems the adapting UNet-based encoder for image classification tasks. From this perspective, a very relevant work is HRNet and its variants which may need to be discussed and compared.
2. The motivation is not very clear. In the title and the contribution summary of the introduction, the authors emphasize lightweight networks. However, the analysis in the introduction is about local and global information extraction.
3. The discussion of related works is not enough. Many recent lightweight networks are not mentioned. In fact, most of the recent networks (e.g., MobileViT, Mobile-Former, MobileOne) achieve a better tradeoff between performance and efficiency compared to the methods in Table 1/2/3. In addition, the variants of self-attention (e.g., [1]) are also needed. In a word, some recent related works need to be discussed and compared.
[1] Fast vision transformers with hilo attention.
4. Experiments: 1. lacking the results on Cifar-10 dataset. 2. More recent methods are required in Table 2/3. 3. The results of ours-3 in Table 2 and Table 3 are not the same. 4. The expression in the text does not match Figure 2, as the worst version is ours-32 in Figure 2, not ours-3. In the ablation study, a comparison of the different data sizes is required if the authors want to prove that the suitable dimension of the network is related to data size. 5. More general datasets, like ImageNet are required for the comparison.
5. Writing and Typos, on Page 7, first paragraph,  H’ and W’. It's better to provide the shape of variables in the section 3.1.1.

**Questions:**

In my opinion, this paper needs to be further improved. The questions and suggestions are listed in the Weaknesses.

---

### Official Review · Reviewer_pv9C · 2023-11-01

**Soundness:** 2 fair
**Presentation:** 2 fair
**Contribution:** 2 fair
**Rating:** 3
**Confidence:** 5

**Summary:**

The authors propose a lightweight network structure based on UNet for small sample classification with only 2.5M parameters. The gradient calculation formula is given to show the structure inherits the gradient calculation advantages of ResNet. The proposed model can handle multi-scale image classification tasks. The interpretability of the model is discussed and an inference is proposed to explain the feature space dimension.

**Strengths:**

1. The paper is easy to follow.
2. The proposed method has small amount of parameters and FLOPs.

**Weaknesses:**

1. The proposed method is lack of novelty since it is a simple combination of Unet, Attention, and residual.
2. The presentation should be re-organized, e.g. figure 1, Table 1, figure 2, etc. There are some typos, e.g. Table 3.
3. I don't quite understand the motivation to choose the Unet for image classification.
4. Some powerful baselines like Mobilenetv3, MixNet, GhostNet, etc. are not compared.
5. I remember the Mobilenet was proposed in 2016. There is something wrong with the reference.

**Questions:**

See weakness.

---

### Official Review · Reviewer_YaL5 · 2023-11-08

**Soundness:** 2 fair
**Presentation:** 2 fair
**Contribution:** 2 fair
**Rating:** 5
**Confidence:** 5

**Summary:**

This paper focuses on lightweighted neural network and combines Resnet with Unet to present a simple ResUnet, where the residual connection is exploited in UNet, and the classifier is designed with attention mechanism. Experiments on some datasets show the superiority and the interpretability is discussed.

**Strengths:**

1. The proposal is simple and the idea is intuitive by combining the adverantage of Resnet and Unet.

2. The writting clear reflects the technical idea of this work.

**Weaknesses:**

1. The novelty is limited. First, the resnet has been used to improve the Unet, i.e., resUnet. Therefore, the idea is not novel. Second, the attention mechanism is a mature idea, and I could not learn more from this work.

2. The technical contribution is incremental and motivation is not strong. A number of lightweighted networks were designed, but there lacks analysis of such networks, and why the proposed method is a better choice. Therefore, stronger motivation for each component in the proposed method is needed.

3. The framework combines existing techniques, and the overall contribution is weak.

4. There lacks more experiments on benchmark datasets such as Imagenet for image classification, a typical vision task. Therefore, this work contributes less on image classification.

**Questions:**

1. The presentation of Fig.4 is too intuitive comparing to Fig.1. Fig.4 seems not a special case of this work. It can also be useful for others. Therefore, this figure can be more specific for interpreting the proposed model.

2. More ablation study is necessary for proving the effectiveness of each component, such as attention classifier.

---

### Meta-Review · Area_Chair_7kdj · 2023-12-03

**Metareview:**

The paper presents a lightweight network, which is a combination of ResNet and Unet, for image classification. It seems that the proposed technique lacks novelty. All reviewers suggest rejection and the authors didn't submit their rebuttal, thus I would recommend rejecting the paper.

**Justification For Why Not Higher Score:**

The paper presents a lightweight network, which is a combination of ResNet and Unet, for image classification. It seems that the proposed technique lacks novelty. All reviewers suggest rejection and the authors didn't submit their rebuttal, thus I would recommend rejecting the paper.

**Justification For Why Not Lower Score:**

N/A

---

### Decision · Program_Chairs · 2024-01-16

Reject